# Personality Variables as Predictors of Health Services Consumption

**DOI:** 10.3390/ijerph18105161

**Published:** 2021-05-13

**Authors:** Antonio Taboada-Vázquez, Ruben Gonzalez-Rodriguez, Manuel Gandoy-Crego, Miguel Clemente

**Affiliations:** 1Department of Psychology, Universidade da Coruna, 15071 A Coruna, Spain; antonio.taboadav@udc.es (A.T.-V.); miguel.clemente@udc.es (M.C.); 2Department of Social Work, Universidade de Vigo, 36310 Vigo, Spain; rubgonzalez@uvigo.es; 3Department of Psychiatry, Radiology and Public Health, Universidad de Santiago de Compostela, 15890 Santiago de Compostela, Spain

**Keywords:** environment and public health, health services, personality, personal health services, public health

## Abstract

Expenditure on healthcare and services can be a serious problem for public health. Personality variables should be included as indicators to be considered when studying the consumption of health resources and their planning. This study aims to identify the psychological and psychosocial variables that identify people who can be considered high consumers of health resources versus those who barely consume such resources. The sample was made up of a total of 1124 subjects; one half were men, and one half were women, all of legal age and residents in Spain. A battery of tests was created that included a questionnaire of sociodemographic variables and of healthcare consumption, as well as several psychological variables (Zimbardo Time Paradox Inventory, Multidimensional Locus of Control Scale, Psychological Reactance Scale, Coping Responses Inventory, self-efficacy scale applied to health, and the Symptom Checklist-90-R). The following variables of the model were significant predictors (*p* ≤ 0.05): a negative past, a fatalistic present, psychological cognitive reactance, behavioral coping, health self-efficacy, and the level of somatization. Data from the statistical analyses show how to create a psychological profile of people who are high consumers of healthcare resources that will allow for the creation of intervention programs in this regard.

## 1. Introduction

Healthcare services are those aimed at the diagnosis and treatment of diseases as well as at the maintenance of health. The provision of health services to the population is itself a public health problem, as it usually involves an expense that is becoming increasingly more difficult for countries to maintain. This paper focuses on the reality of the existence of certain groups that consume more health services (general or specialist consultations, surgery, diagnostic tests, and pharmaceutical expenditure), whereas others consume fewer. Similarly, due to their personality traits, some people are more likely to consume health services and, contrariwise, others consume less. The determination of the psychological typology that implies higher or lower consumption of health services can help to create mechanisms that, without undermining healthcare, would lead to savings in such assistance; for example, through collective campaigns and individual training to promote the existence of a profile of a low consumer of health services.

We are aware that the use of health services does not only depend on the psychological variables that the users of the system have. We believe that the choice of a type of health system by each government has a determining weight and that healthcare should be organized according to the demographic weight of each group and their respective healthcare needs. Thus, every system must be able to specially attend to various groups, including older people, disabled people, or those with mental health problems. This work, given its orientation, only focuses on the psychological part of the problem, an issue that, in our opinion, is little investigated.

Various studies have linked the consumption of health services to variables inherent to human development, such as age, physical or mental health status, destructive over-dependence (DO), or dysfunctional detachment (DD) [1,2,3,4].

Our interest is focused on the determination of the psychological predictor variables that can explain higher or lower consumption of resources. One of the most commonly investigated aspects is the pattern of behavior, which studies the person’s vulnerability to surrounding stressors and the short and long-term consequences for health. Cardiologists Friedman and Rosenman [5] identified the Type A behavioral pattern, typical of young men with ischemic heart disease. The appropriate intervention would be to try to help the person to evolve from acute Type A behavior to Type B, as subjects with a Type B personality pattern consume far fewer health resources. Subsequently, Morris and Greer [6] labelled the Type C personality, characterized by emotional inhibition; diseases associated with this personality type are rheumatism, infections, allergies, skin disorders, and cancer. Finally, Denollet and Brutsaert [7] labelled the Type D (distress) personality pattern. It is also characterized by inhibition, but in this case of a social nature, not emotional. A Type D personality predisposes one to anxiety and depression and is a predictor of long-term mortality from cardiovascular disorder. There are multiple studies that relate a Type D personality pattern to increased consumption of health resources [8,9,10].

Another variable studied in the relationship between people’s health and their use of healthcare resources is the perception of self-efficacy. This variable [11] is considered an excellent predictor of many behaviors [12]. In general, research has verified that individuals with high perceived self-efficacy are more likely to perform preventive behaviors, to seek treatments earlier, and to be more optimistic about their effects [13,14,15,16]. A specific instrument (the Self-Efficacy Health [SEH] scale) that measures self-efficacy in situations of health has even been created [17]. The relation of healthcare consumption with personality variables has also been studied, especially with the Zimbardo Time Paradox Inventory [18], Derogatis’ Symptom Check-List [19,20,21,22], through the locus of control or internalization-externalization [23,24,25,26], as well as through psychological reactance [27,28,29,30,31], or coping [32,33,34].

This research follows the work of Schoormans et al. [35], who pointed out that, in some cases, the use of healthcare is not only determined by the complexity of the health process or by patients’ functional status, but also by their psychological characteristics. Kessler and Maclean [36] found associations between the Big Five test of personality factors and measures of alcohol consumption and abuse, which is an important cardiovascular risk factor. Powers et al. [37] analyzed people with personality pathologies, verifying that they are an important predictor of increased consumption of health resources.

We think that personality variables should be included as indicators to be considered when studying the consumption and planning of health resources. For this reason, the present study aims to identify which psychological and psychosocial variables allow us to identify people who can be considered high consumers of health resources compared to those who barely consume such resources, in periods when there is no existence of a sudden severe disease, a cognitive decline due to age, or a situation of age-related dependence. We hypothesize that a differentiating profile can be created in which variables related to self-efficacy and an optimistic view of the world will load more on the profile of the low consumer of health resources, and in contrast, people with a lower degree of self-efficacy regarding health problems and who are more pessimistic about life will be higher consumers of health resources.

Access to public health systems must be free, universal, and independent of the personal characteristics of each user. However, we believe that from the point of view of scientific knowledge, it is good to determine which personality variables are manifested by those who use the health services the most and which are manifested by those who use the system the least. The aim is to avoid that the professionals who serve users use knowledge based solely on experience and can instead use scientific knowledge in this regard.

## 2. Materials and Methods

### 2.1. Participants

The sample was made up of a total of 1124 people, of whom one half were men and one half were women. All were 18 years or older, with an average age of 37.53 years (SD = 17.91; range 18 to 65 years). All participants lived in the region of Galicia (Spain) and were located in their homes, used for their recruitment surveyors who were enrolled in university studies in psychology and nursing. We explained to the participants the reason for the study and all of them signed an informed consent form.

Since we are all users of health services, the population is very large. In the case of Spain, the population is over 47 million, about 46.5 million of whom are legal age (18 years or more, that is, those selected in the study). For this reason, the criterion of exceeding 1111 subjects in the sample was used. In any case, the coefficient d was calculated to determine the effect size, obtaining a *t*-test value of 0.81 (very suitable).

### 2.2. Scales Used

We created a battery of tests that included the following questionnaires:

A questionnaire of sociodemographic variables and healthcare consumption. Regarding the sociodemographic variables, we considered age, sex, level of completed studies, and marital status. The variables concerning healthcare consumption (always within the last two months) were as follows: the number of times that the person went to the medical practitioner; the number of times that the person consulted a specialist; the number of diagnostic or analytical tests carried out; and, if the person went to a rehabilitation service, the number of sessions attended.

The Zimbardo Time Paradox Inventory [18]. In this study, we used the Spanish version of Diaz-Morales [38], although it was adapted for other populations [39,40,41]. The perception of time represents an essential element of cognition, given that the experiences that people live through daily have significance, regulating their behavior and impacting the future. The present of the subjects is associated with the experiences lived in the past, and it is understood that the present will be to a greater or lesser extent linked to what is projected in the future and to the expectations and goals that each subject intends to achieve. That is, what the subjects decide to do in the present will be associated both with the experiences they have already lived as well as the future they project.

The Multidimensional Locus of Control Scale (internality-externality, I-E) [42]. We used the Spanish version of Romero-García [43]. The variable locus of control, or internal-external control of reinforcement, has been one of the most common in social psychology. The construct is part of Rotter’s theory of social learning and refers to how people explain a posteriori the events that happen to them either internally (they are responsible) or externally (they are not responsible, but everything is due to external forces such as the environment, others, etc.).

The Psychological Reactance Scale (RP) [44] based on Brehm’s theory of reactance [45] in the version translated by Perez [46] considers two dimensions: the affective component and the cognitive component. The theory of psychological reactance proposes that reactance is a motivational force that is activated when perceived behavioral freedoms are eliminated or threatened with elimination. This motivation is aimed at restoring those freedoms and can be expressed through a number of direct and indirect ways.

The Coping Responses Inventory (CRI) [47,48], in the adaptation of Ongarato et al. [49]. Many are the instruments created from psychology to evaluate coping strategies in the face of stress, majority of them derived from the theory of Lazarus and Folkman (1984). We have used one of the most common.

A scale of self-efficacy applied to health (SEH) based on the test of self-efficacy of Baessler and Schwarzer [50], but with the items referring only to issues about the health status. The scale consisted of 10 items, rated on a four-point Likert format: strongly disagree to strongly agree. Reliability studies of this scale yielded Cronbach alpha values of 0.771 and reported positive results concerning its validity [17]. The SEH is a brief instrument that allows for the determination of individuals with high levels of coping with health problems that may arise.

The Symptom Checklist-90-R [19,20,21,22] is an instrument that was developed to assess and quantify individuals’ patterns of symptoms, which can be used both in community and clinical diagnostic tasks. The Symptom Checklist-90-R (SCL-90-R) is a symptomatic quantification instrument designed at Johns Hopkins University that allows the evaluation of a wide range of psychological and psychopathological symptoms. It can assess the presence of a lot of symptoms and determine their intensity.

### 2.3. Procedure

The questionnaires were applied individually in the homes of the participants. After gathering the information, the database was created with the statistical program SPSS version 22, and after reviewing it, the data were subjected to the following relevant statistical tests: (1) we used Cronbach’s alpha to determine the reliability of all the tests; (2) we determined the normality of the variables in order to choose the most appropriate regression technique. This was calculated with the Chi-square test; (3) we determined a global rate of consumption of healthcare resources composed of the total sum of the values that were collected in the corresponding questionnaire; and (4) we determined the regression, using the greater or lesser use of health resources as the criterion variable, with all of the personality variables as predictors. The criterion variable was dichotomized (lesser consumption of health resources vs. greater consumption); participants whose level of healthcare consumption was in the first tercile made up the first group, and participants with a higher level of consumption of resources (third tercile) formed the second group. Participants who obtained intermediate scores (second tercile) were eliminated from the analyses.

We explained the research to each participant individually, and all of them gave their signed informed consent. None of them refused to participate. We also previously requested authorization to conduct the research from the Ethics Committee of the University of Coruna, receiving a positive report. This research respects the principles laid down by the Declaration of Helsinki.

## 3. Results

All the applied questionnaires presented fully acceptable levels of reliability (between 0.81 and 0.99). In addition, all the predictor variables showed a satisfactory fit to the normal curve. As a result, we decided to use a statistical technique of binary logistic regression. The estimation of the model ended at the fourth iteration, because the estimates of the parameters changed less than 0.001. The likelihood value of −2 log was 1123.258; Cox and Snell’s R^2^ was 0.37; and Nagelkerke’s R^2^ was 0.183. The classification table shows an improvement of 14.6%.

The significantly predictive variables (*p* ≤ 0.05) of the model were as follows: a negative past (positively, i.e., predicting high consumption of health services); a fatalistic present (positively); psychological cognitive reactance (negatively, i.e., subjects with low cognitive reactance consume more health resources); behavioral coping (positively, but it should be taken into account that higher scores on the scale imply larger coping deficits, i.e., lower levels of behavioral coping correspond to higher consumption of health services); health self-efficacy (negatively, i.e., lower levels of health self-efficacy imply increased consumption of health resources); and somatization level (higher level of somatization corresponds to increased consumption of health resources). These results can be found in Table 1, which presents both the significant and nonsignificant predictive variables as well as main statistics of each variable.

## 4. Conclusions

Data from the statistical analyses show how to create a psychological profile of high consumers of health resources, thereby confirming the proposed first hypothesis.

On the other hand, the obtained profile is consistent with the investigations to date. We confirmed that Zimbardo and Derogatis’ questionnaires are both suitable to detect personality variables that can predict the consumption of health services. Two variables from Zimbardo’s questionnaire load on the profile: a negative past and a fatalistic present. Out of the five variables contemplated by the instrument, these are negative; that is, people who tend to perceive and assess their past negatively, or to be fatalistic in the interpretation of their present consume more healthcare resources. With regard to Derogatis’ questionnaire, the somatic level becomes a key explanatory variable (people with higher levels of somatization consume more healthcare). It should be noted, however, that variables clearly identified in the literature as predictors, such as depression or anxiety, are not significant when creating the profile.

The hypothesis regarding self-efficacy is also confirmed. The Health Self-Efficacy (SEH) scale of Gandoy-Crego et al. [17] shows that subjects with a lower perception of knowing how to face possible health problems are major consumers of health resources.

Two variables of minor importance in the literature, psychological reactance and coping, also adequately predict the level of consumption of health services. However, it is necessary to note that regarding reactance, the variable that enters the equation is cognitive reactance (individuals who consider disobeying doctors’ recommendations when they have a health problem, and the affective value of such disobedience is irrelevant), and that regarding coping, only behavioral coping is significant (people who are unwilling to deal with a health problem, and therefore unwilling to act accordingly). That is, people who face health problems negatively (they disobey health professionals’ instructions and do not deal with overcoming the problem, eventually obtaining what they do not desire) become high consumers of the health system. The initial hypotheses are confirmed in these two variables, but not in the case of cognitive coping or emotional reactance.

This paper presents a series of limitations that open the door to future research. For example, we note the inclusion of variables that refer to behavioral patterns. These variables were not included as behavioral patterns refer to very specific diseases, and their types are closely related to heart problems. However, future works could investigate whether there is a personality pattern of a “disobedient” person from the point of view of health, and how this would impact the consumption of health services. This idea is consistent with the work of Powers et al. [37], who considered personality pathologies as important predictors of increased consumption of medical resources at advanced ages, proposing their consideration as a risk factor to take into account in order to reduce the overuse of health resources among older adults.

We want to point out that although personality variables are stable, they can be modified through training programs. Thus, this work allows for the preparation of treatment programs for people who are high users of health services, in order to empower populations to deal more actively with their health problems. This would benefit them and society in general by lowering health costs. No doubt, as already noted by Kessler and Maclean [36] and Schoormans et al. [35] among others, personality variables should be included among the indicators that allow for the increase in the quality of the population’s health and reducing expenditure on health. In summation, there are a number of advantages. The first is the advancement of science to determine the profile of people who are high consumers of health services versus those who are not, regardless of their respective health status. The second is the power to create programs that, without barring access to the health system for all citizens, prevent its use when it is not necessary. Finally, the power to create protocols for health professionals that allow them to distinguish in a more adequate way if when a person demands an unnecessary test or medication, it may be due to his personality type.

We want to remember what was already expressed in the Introduction of this work. The variables with the greatest weight when determining the use of the health system depend on the organization of each government and their ability to ensure healthcare for all citizens, especially for groups that require further assistance. In any case we consider that this work, by focusing on psychological variables, allows us to complete the study of the factors that explain that within the same groups, some people use these services excessively, while others hardly use them.

On the other hand, we consider that future studies should take into consideration aspects such as the diagnosis of each disease, the time elapsed when verifying said diagnosis through medical tests, the treatment followed, and the evaluation after possible treatments.

## Figures and Tables

**Table 1 ijerph-18-05161-t001:** Personality variables as predictors of consumption of health services.

Predictor Variable	B	SE	Wald	df	*p*	Exp(B)
Negative Past	0.446	0.164	7.352	1	0.007	1.562
Hedonist present	0.104	0.169	0.379	1	0.538	1.109
Future	0.204	0.169	1.443	1	0.230	1.226
Positive past	−0.022	0.169	0.017	1	0.897	0.978
Fatalistic present	0.450	0.168	7.214	1	0.007	1.568
Internal control	0.278	0.231	1.450	1	0.229	1.321
External control by chance	0.016	0.227	0.005	1	0.945	1.016
External control by power	−0.055	0.204	0.072	1	0.788	0.947
Emotional reactance	0.026	0.188	0.020	1	0.889	1.027
Cognitive reactance	−0.378	0.185	4.178	1	0.041	0.685
Cognitive coping	0.140	0.174	0.651	1	0.420	1.150
Coping behavior	0.306	0.140	4.793	1	0.029	1.358
Cognitive avoidance coping	−0.274	0.152	3.271	1	0.070	0.760
Behavioral avoidance coping	0.035	0.137	0.066	1	0.797	1.036
Health self-efficacy	−0.925	0.221	17.479	1	0.000	0.397
Somatizations	0.780	0.179	19.033	1	0.000	2.181
Obsession-compulsion	−0.213	0.189	1.281	1	0.258	0.808
Interpersonal Sensitivity	−0.167	0.204	0.669	1	0.413	0.846
Depression	0.091	0.124	0.540	1	0.462	1.096
Anxiety	−0.022	0.118	0.034	1	0.853	0.978
Hostility	0.052	0.151	0.120	1	0.729	1.054
Phobic Anxiety	0.315	0.219	2.069	1	0.150	1.370
Paranoid Ideation	−0.318	0.180	3.112	1	0.078	0.728
Psychoticism	0.143	0.255	0.316	1	0.574	1.154
Constant	−2.053	1.151	3.183	1	0.074	0.128

Note: SE: Standard error; B: Beta coefficient; df: degrees of freedom; sign: significance.

## Data Availability

http://dx.doi.org/10.17632/vyhjbns9kn.1 (accessed on 12 May 2021) and can also be requested from the corresponding author.

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
