# Peer review of "Personality Variables as Predictors of Health Services Consumption"

_ijerph, 2021, doi:10.3390/ijerph18105161_

Round 1
Reviewer 1 Report
They address a topic of interest, with a large sample, and interesting conclusions can be drawn from the results regarding their applicability. Therefore, my congratulations in this regard.
However, like all scientific work, yours can also be improved. Therefore, in the following paragraphs I will make a series of considerations that I believe could improve the communication of your results and the scope of their applicability. These affect the objective, method and discussion of your results.
The predictor variables are defined operationally, that is, according to the concept underlying the instrument used to measure them. The result may be of applied interest, since there is no doubt that knowing the personality characteristics that predict the consumption of health services could be used to design interventions that raise the awareness of the population and health professionals in this respect, that increase people's capacity to face illness and pain, that reduce the reaction to following health prescriptions, that allow patients to be screened according to their fatalistic expectations and their level of somatization, offering in these cases greater and better coverage in mental health, since this is usually insufficient and generalist in the public health systems. In this sense, they should make an effort (especially in the discussion of the results) and propose concrete applications derived from the results obtained, especially when the general proposal they make gives the impression of blaming the citizen for the abusive use of health services, ignoring that the excessive or, perhaps, unnecessary use of health services does not only depend on personal disposition, but also on the way in which they are administered and managed (e.g., insufficient health services to meet the needs of the population, insufficient health services to deal with mental health problems or inadequacy to deal more effectively with elderly people with multiple pathologies). Likewise, they should also consider the legal margins of their proposals, since access to public health systems is free and cannot be subjected to conditions or personal characteristics.
The sample size is large and probably representative of the general population from which it was drawn, but they do not provide information that would allow it to be accurately assessed; that is, they should report on the design and sampling procedures employed, as well as on the power and precision of the sample.
The results themselves are valid; that is, they have been obtained through the application of reliable and valid instruments, coincide in part with the findings of previous research and could be applicable, as mentioned above, provided they clarify the sampling issues raised and deepen the applicability of the same, taking into account possible confounding factors (eg, mental health and geriatric coverage) and proposing applications that could be integrated into the public health system without the need to open a debate on their regulatory feasibility.
Author Response
|
Reviewer # 1
|
Answers |
|
They should make an effort (especially in the discussion of the results) and propose concrete applications derived from the results obtained, especially when the general proposal they make gives the impression of blaming the citizen for the abusive use of health services, ignoring that the excessive or, perhaps, unnecessary use of health services does not only depend on personal disposition, but also on how they are administered and managed (e.g., insufficient health services to meet the needs of the population, insufficient health services to deal with mental health problems or inadequacy to deal more effectively with elderly people with multiple pathologies). |
Thanks for your congratulations. We appreciate your comment. Even though, as reflected in the title of the manuscript, we intend to determine the importance of psychological variables, it seems very appropriate to reflect that the problem exceeds the psychological scope. In this way, we have added a text in the Introduction (in the second of the paragraphs, page 1), and in the last paragraph of Discussion and Conclusions, as a closing of the manuscript.
In “Introduction”: “We are aware that the use of health services does not only depend on the psychological variables that the users of the system have. We believe that the choice of a type of health system by each government has a determining weight and that health care should be organized according to the weight of each group according to their health care needs. Thus, every system must be able to attend especially to the group of older people, disabled people, or those with people with mental health problems. This work, given its orientation, only focuses on the psychological part of the problem, an issue, in our opinion, little investigated. "
In “Discussion and Conclusions”: “We want to remember what was already expressed in the Introduction of this work. The variables with the greatest weight when determining the use of the health system depend on the organization carried out by each government, and such management must be such as to ensure health care for all citizens, and especially those groups that require further assistance. In any case, we consider that this work, by focusing on psychological variables, allows us to complete the study of the factors that explain that within the same groups, some people use these services excessively, and others hardly use them. " |
|
Likewise, they should also consider the legal margins of their proposals, since access to public health systems is free and cannot be subjected to conditions or personal characteristics. |
We totally agree with you. Access to public health systems must be free and universal and must be independent of the personal characteristics of each user. However, we believe that from the point of view of scientific knowledge it is good to determine which personality variables are manifested by those who use the health services the most, and which are those who use the system the least. The aim is to avoid that the professionals who serve users use knowledge-based solely on experience, and can use scientific knowledge in this regard. This is how we reflect it by adding the following paragraph at the end of “Introduction”:
“Access to public health systems must be free and universal and must be independent of the personal characteristics of each user. However, we believe that from the point of view of scientific knowledge it is good to determine which personality variables are manifested by those who use the health services the most, and which are those who use the system the least. The aim is to avoid that the professionals who serve users use knowledge-based solely on experience, and can use scientific knowledge in this regard. " |
|
The sample size is large and probably representative of the general population from which it was drawn, but they do not provide information that would allow it to be accurately assessed; that is, they should report on the design and sampling procedures employed, as well as on the power and precision of the sample. |
We totally agree with you. The following paragraph is added in Participants, don de informal in addition to the effect size:
“Since we are all users of health services, the population is very large. In the Spanish case, the number of people is more than 47 million, of whom of legal age (18 years or more, that is, those selected in the study) are about 46.5 million. For this reason, the criterion of exceeding 1,111 subjects in the sample was used. In any case, the coefficient d was calculated to determine the effect size, obtaining a T-test value of .81, that is, large. " |
|
The results themselves are valid; that is, they have been obtained through the application of reliable and valid instruments, coincide in part with the findings of previous research and could be applicable, as mentioned above, provided they clarify the sampling issues raised and deepen the applicability of the same, taking into account possible confounding factors (eg, mental health and geriatric coverage) and proposing applications that could be integrated into the public health system without the need to open a debate on their regulatory feasibility. |
Thank you very much for your comment. We believe that the first of the questions have already been considered, thanks to your previous questions.
Regarding applications, the following sentence has been added in Discussion and Conclusions, after the limitations:
“In sum, there are several advantages. The first is the advance that science supposes to determine the profile of those people who are high consumers of health services versus those who are not, regardless of their health as such. The second, the power to create programs that, without questioning access to the health system for all citizens, avoid using it when it is not necessary. And finally, the power to create protocols for health professionals that allow them to distinguish more adequately if, when a person demands an unnecessary test or medication, it may be due to his personality type.” |

Reviewer 2 Report
Personality variables as predictors of health services consumption
In summary, the authors declared that the aim of the work is:
- creating a psychological profile of people who are more likely to benefit from healthcare.
- guide how to design a training programme for the population to achieve savings in health services.
Unfortunately, the authors did not provide a profile of people who are more likely to use healthcare. What is more, they did not indicate how to design a training program to achieve savings in benefits.
The summary does not indicate which tools were used to verify research problems. No period and results of the study were given. The summary should be improved. I suggest indicating the purpose of the study, methodology and conclusions.
The introduction begins with a repetition of the summary. Personality patterns were presented using relatively modest literature. In the bibliography, there is no novelty from the literature of the subject. At the end of the introduction, it was assumed that people are pessimistic about using healthcare resources more often.
The methods of the study are described chaotically. Only the names of the tools used are listed.
The authors did not select the appropriate methods to analyse the collected data for the intended purpose. The results obtained have not been correctly interpreted.
The article requires a substantial correction and, in principle, to write from the beginning.
Author Response
|
In summary, the authors declared that the aim of the work is:
ü Creating a psychological profile of people who are more likely to benefit from healthcare. ü Guide how to design a training program for the population to achieve savings in health services. Unfortunately, the authors did not provide a profile of people who are more likely to use healthcare. What is more, they did not indicate how to design a training program to achieve savings in benefits. |
Thank you very much for your comments. We believe that you may have thought that the profile we wanted to create was sociological or demographic, while our profile is psychological. The profile that we have created is shown in Table 1. Regarding the possible training program, our intention is not to create it, but rather that such a program can be designed based on the results of this research. That is why the title of the manuscript is “Personality variables as predictors of health services consumption”. |
|
The summary does not indicate which tools were used to verify research problems. No period and results of the study were given. The summary should be improved. I suggest indicating the purpose of the study, methodology, and conclusions. |
Thank you very much for your observation. We have created a new abstract taking into account your suggestions and the publication rules, which specify that it should not exceed 200 words. We have added what psychological tests we have used, and modified the end of the paragraph to make it clear that we did not create an intervention program. The new Abstract is as follows:
“Expenditure on health-care and services can be a serious problem for public health. Personality variables should be included as indicators to be considered when studying the consumption of health resources and their planning. This study aims to identify the psychological and psycho-social variables that identify people who can be considered high consumers of health resources versus those who barely consume such resources. The sample was made up of a total of 1124 subjects; one half were men and one-half women, all of legal age and residents in Spain. A battery of tests was created that included a questionnaire of sociodemographic variables and health-care consumption, as well as several psychological variables (Zimbardo Time Paradox Inventory, Multidimensional Locus of Control Scale, Psychological Reactance Scale, Coping Responses Inventory, Self-efficacy scale applied to health, and the Symptom Check-List-90-R). The following variables of the model were significant predictors (p ≤ .05): a negative past, a fatalistic present, psychological cognitive reactance, behavioral coping, health self-efficacy, and the level of somatization. Data from the statistical analyses show how to create a psychological profile of people who are high consumers of health-care resources they will allow the creation of intervention programs in this regard.” |
|
The introduction begins with a repetition of the summary. Personality patterns were presented using relatively modest literature. In the bibliography, there is no novelty from the literature of the subject. At the end of the introduction, it was assumed that people are pessimistic about using healthcare resources more often. |
Thank you very much for your comments. We have reviewed the beginning of the “Introduction” section but we have not wanted to eliminate anything since we consider that it is more indicative for the future reader not to eliminate text. Regarding bibliographic references, we would appreciate your guidance in indicating newer works. The end of the Introduction, as usual in the scientific works of Psychology, expresses the hypotheses of our work. |
|
The methods of the study are described chaotically. Only the names of the tools used are listed. |
To build the Methodology section we have followed the rules of the APA (American Psychological Association).
|
|
The authors did not select the appropriate methods to analyze the collected data for the intended purpose. The results obtained have not been correctly interpreted. |
We respect your comment, but we do not understand it, you do not tell us where the errors are. We reiterate that we strictly follow the APA standards. |

Reviewer 3 Report
While this is good work and somewhat novel in its approach, it is not without its biases.
The first of these is the relationship between diagnosis, time to diagnosis, complications and/or exacerbations and consumption of health care resources.
This is not sufficiently explained.
On the other hand, the predictive model lacks parsimony, which is essential when trying to explain the influence of one variable (resource consumption) with other variables.
The interaction factors between variables have not been analysed.
Author Response
|
The relationship between diagnosis, time to diagnosis, complications and/or exacerbations, and consumption of health care resources. This is not sufficiently explained. |
Thank you very much for your observation. In fact, we have not taken into account the aspects that you indicate to us, since it was impossible for us to consider so many variables and analyze them. We believe that these issues should be considered for future studies, so we have added this aspect in the limitations, at the end of the manuscript:
"On the other hand, we consider that future studies should take into consideration aspects such as the diagnosis of each disease, the time elapsed when verifying said diagnosis through medical tests, the treatment followed, and the evaluation after possible treatments." |
|
On the other hand, the predictive model lacks parsimony, which is essential when trying to explain the influence of one variable (resource consumption) with other variables. |
We regret that we do not know of other statistical techniques that respect this principle more adequately. The mathematical model of logistic regression is based on adding one by one all the variables that are provided. We understand that the principle of parsimony is met, but we would appreciate your suggesting other alternatives. |
|
The interaction factors between variables have not been analysed. |
We consider that the interaction between the variables is considered since we use the statistical technique of regression. This technique considers the interaction between all the variables that are introduced in the model and determines the relative weight of each one of them concerning the others. |

Round 2
Reviewer 2 Report
The authors have significantly improved this manuscript based on reviewers' suggestions and I now consider it suitable for publication.
Author Response
Reviewer # 2: Comment: “The authors have significantly improved this manuscript based on reviewers' suggestions and I now consider it suitable for publication”.
Thank you very much. The manuscript has improved a lot thanks to your contributions.

Reviewer 3 Report
No comments on the methodology or the results, although I would like the authors to explain a little better what each of the items consists of, especially those that turned out to be relevant (negative past, fatalistic present, psychological cognitive reactance, behavioural coping, health self-efficacy, and the level of somatization, in order to provide better information to decision-makers (including clinicians).
Author Response
Dear Reviewer:
Your comment: “Comment: “No comments on the methodology or the results, although I would like the authors to explain a little better what each of the items consists of, especially those that turned out to be relevant (negative past, fatalistic present, psychological cognitive reactance, behavioural coping, health self-efficacy, and the level of somatization, in order to provide better information to decision-makers (including clinicians).”
Thank you very much about yours observations. We are added information about each instrument in seccion 2.2. (Scales used). The text added is:
The perception of time represents an essential element of cognition, given that the experiences that people live daily have significance, regulating their behavior and anticipating the future. The present of the subjects is associated with the experiences lived in the past, as well as it is understood that the present will be to a greater or lesser extent linked to what is projected in the future, to the expectations and goals that each subject intends to achieve. That is, what the subjects decide to do in the present will be associated with the experiences already lived, as well as the future they project.
The variable locus of control, or internal-external control of reinforcement, has been one of the most common in Social Psychology. The construct is part of Rotter's theory of social learning and refers to how people explain "a posteriori" the events that happen to them, either internally (they are responsible) or externally (they are not responsible, but everything is due to external forces such as the environment, others, etc.)
The theory of psychological reactance proposes that reactance is a motivational force that is activated when perceived behavioral freedoms are eliminated, or threatened with elimination. This motivation is aimed at restoring those freedoms and can be expressed through a number of direct and indirect ways.
Many are the instruments created from Psychology to evaluate coping strategies in the face of stress, mostly, all of them derived from the theory of Lazarus and Folkman (1984). We have used one of the most common.
The SEH is a brief instrument that allows determining if individuals have high levels of coping with health problems that may arise.
The Symptom Checklist-90-R (SCL-90-R) is a symptomatic quantification instrument designed at Johns Hopkins University that allows the evaluation of a wide range of psychological and psychopathological symptoms. Assess the presence of 90 symptoms and determine their intensity.
Thank you very much for your support.
